# MS-GS: Multi-Appearance Sparse-View 3D Gaussian Splatting in the Wild

**Deming Li**
Johns Hopkins University
Baltimore, MD 21218
dli90@jhu.edu

**Kaiwen Jiang**
University of California, San Diego
La Jolla, CA 92093
k1jiang@ucsd.edu

**Yutao Tang**
Johns Hopkins University
Baltimore, MD 21218
ytang67@jhu.edu

**Ravi Ramamoorthi**
University of California, San Diego
La Jolla, CA 92093
ravir@ucsd.edu

**Rama Chellappa**
Johns Hopkins University
Baltimore, MD 21218
rchella4@jhu.edu

**Cheng Peng**
Johns Hopkins University
Baltimore, MD 21218
cpeng26@jhu.edu

## Abstract

In-the-wild photo collections often contain limited volumes of imagery and exhibit multiple appearances, e.g., taken at different times of day or seasons, posing significant challenges to scene reconstruction and novel view synthesis. Although recent adaptations of Neural Radiance Field (NeRF) and 3D Gaussian Splatting (3DGS) have improved in these areas, they tend to oversmooth and are prone to overfitting. In this paper, we present MS-GS, a novel framework designed with **M**ulti-appearance capabilities in **S**parse-view scenarios using 3D**GS**. To address the lack of support due to sparse initializations, our approach is built on the geometric priors elicited from monocular depth estimations. The key lies in extracting and utilizing local semantic regions with a Structure-from-Motion (SfM) points anchored algorithm for reliable alignment and geometry cues. Then, to introduce multi-view constraints, we propose a series of geometry-guided supervision steps at virtual views in pixel and feature levels to encourage 3D consistency and reduce overfitting. We also introduce a dataset and an in-the-wild experiment setting to set up more realistic benchmarks. We demonstrate that MS-GS achieves photorealistic renderings under various challenging sparse-view and multi-appearance conditions, and outperforms existing approaches significantly across different datasets.

## 1 Introduction

High-quality scene reconstruction and novel view synthesis from images is a long-standing research problem with wide-ranging applications in AR/VR, 3D site modeling, autonomous driving, robotics, etc. Remarkably, neural radiance field (NeRF) [1] and 3D Gaussian Splatting (3DGS) [2] achieve photorealistic novel view synthesis with differentiable rendering pipelines and different scene parameterizations. Both approaches build on the fundamental constraint that a voxel in space projects to similar photometric values across views, a constraint that requires a dense scene coverage and multi-view consistency. In practice, such coverage and consistency are often not guaranteed, largely affecting their performance in unconstrained settings.

39th Conference on Neural Information Processing Systems (NeurIPS 2025).

Figure 1: With **20 input views**, DNGS and FSGS produce overly smooth rendering in regions lacking support from sparse point cloud initialization. For scenes with multiple appearances and sparse inputs, methods like GS-W and Wild-GS experience large artifacts at novel views. In contrast, our method in Fig. 1d and 1h renders details and provides a coherent reconstruction.

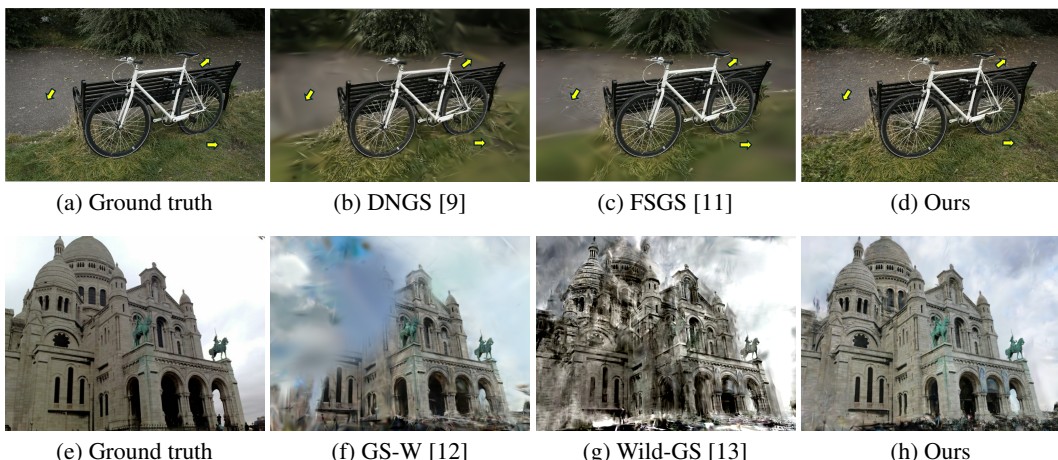

|                     |                 |                  |            |
| (a) Ground truth    | (b) DNGS [9]    | (c) FSGS [11]    | (d) Ours   |
| (e) Ground truth    | (f) GS-W [12]   | (g) Wild-GS [13] | (h) Ours   |

With sparse image sets, overfitting the photometric objectives to an incorrect geometry is a common issue in novel view synthesis. To counter this, semantic constraints [3], depth and novel-view regularization [4, 5, 6], frequency regularization [7], and ray-entropy minimization [8], have been introduced. While effective, these NeRF-based methods incur a heavy computational cost because volumetric rendering requires densely sampling points along camera rays. More recently, 3DGS adaptations have pushed sparse-view synthesis further by exploiting their explicit representation and fast rasterization. Depth regularization [9], floater pruning [10], and proximity-based Gaussian densification [11] regularize the reconstruction and suppress artifacts during training. Despite improved metrics, these methods remain limited by the standard initialization of a sparse Structure-from-Motion (SfM) point cloud when few features are triangulated correctly. Furthermore, applying monocular depth constraints globally to 3DGS is often inaccurate, leading to noisy gradients that prevent proper densification in sparse regions. As shown in Fig. 1, they synthesize overly smooth regions, while our method recovers fine details.

Beyond limited viewpoints, in-the-wild image sets often exhibit photometric inconsistencies across views. These range from subtle exposure shifts to pronounced appearance changes when the same scene is captured at different times of day or in different weather. NeRF-in-the-Wild [14] first demonstrated the ability to model a canonical 3D structure from multi-appearance imagery; since then, various works have followed up to improve synthesis quality [15, 16] and incorporate this capability to 3DGS-based approaches [12, 13, 17]. These methods require more data to disambiguate image-specific radiance compared to appearance-consistent scenarios. As shown in Fig. 1, a moderate number of views still leads to noisy rendering, greatly limiting the application of these methods.

In this paper, we present MS-GS, which improves the robustness of 3DGS in dealing with unconstrained images when limited viewpoints and varying appearances exist, which is underexplored. We find that the performance of 3DGS relies heavily on the initial point cloud: these explicit structures steer the adaptive control of Gaussians and subsequent optimization. To overcome the limitation of the sparse SfM point cloud with limited views, we draw knowledge from the monocular depth estimators [18, 19, 20] that have rapidly progressed. MS-GS aligns the depth prediction with SfM depth, then back-projects pixels into the scene space for a dense point cloud. A key challenge is that monocular depth estimation is often incorrect at relative depth between objects due to single-view ambiguity. We address this with a **Semantic Depth Alignment** approach. A point-prompted segmentation model [21] is leveraged to extract semantically consistent regions using projected SfM points. We design an iterative refinement algorithm to identify each region–expanding or discarding according to the number of enclosed SfM points–and perform alignment inside it before back-projection. The resulting point cloud is denser and better structured than the original sparse SfM output, helping regularize 3DGS structures and promote Gaussian densification.

To enable sparse-view multi-appearance scene modeling, MS-GS decomposes appearance into an image-specific and Gaussian-specific component. The per-image appearance embedding captures the global appearance variations, while each Gaussian's feature embedding encodes the canonical scene appearance. Under sparse-view conditions, it becomes challenging to disambiguate the image-specific radiance from a consistent scene appearance, resulting in overfitting to training image appearance. Building upon the sufficiently accurate geometry that 3DGS optimizes with our dense initialization, we exploit the use of virtual views for **multi-view** constraints, where a series of **Geometry-guided Supervisions** based on 3D warping is proposed. Specifically, we back-project training images to 3D and then project to virtual views created between training cameras to establish correspondences. Appearance consistency is enforced at pixel and feature levels for precise supervision and handling occluded areas. This approach aims to transfer the well-rendered appearance of training images to multiple views. Coupled with our densified point cloud, this design markedly improves geometric coherence and the rendering quality given sparse and multi-appearance imagery.

In addition, the benchmark dataset Phototoursim [22] is collected through the internet such that each image has a unique appearance. Therefore, methods evaluated on this dataset require access to the test view image to obtain appearance, which is not ideal for train-test separation. To this end, we introduce an unbounded drone dataset that features *multi-view* appearance. By relating to camera metadata, not the pixel information, testing views are rendered with the appearance of training images during evaluation. Similarly, the experimental setting of novel view synthesis in the wild is non-trivial. While methods often assume camera poses are exact, sparse-view and multi-appearance registration are themselves prone to error. Accurate reflection of performance in the wild needs to account for realistic registration noise, especially when the underlying SfM point cloud is the input to the 3DGS-based methods. We therefore advocate a protocol that disentangles training and testing cameras during registration and preserves real-world pose uncertainties.

In summary, the main contributions of our work are:

- We introduce a Semantic Depth Alignment approach, which leverages monocular depths in local semantic regions to construct a dense point cloud initialization and significantly improves fidelity in regions with limited overlap.
- We propose a series of Multi-view Geometry-guided Supervision steps based on 3D warping at pixel and feature levels; such a framework reduces overfitting to limited observation and encourages 3D geometry and appearance consistency.
- We evaluate our overall method, MS-GS, across various benchmark datasets in different evaluation settings. MS-GS demonstrates significant quantitative and visual improvements compared to SoTA methods.

## 2    Related Work

**Sparse-view Novel View Synthesis**    To improve sparse-view novel view synthesis, DietNeRF [3] uses scene semantics from a pre-trained visual encoder to constrain a 3D representation. RegNeRF [4] regularizes geometry by enforcing smoothness on rendered depth and appearance by a normalizing flow model in patches from unseen viewpoints. FreeNeRF [7] proposes a low-to-high frequency schedule and penalization on density fields near the camera as regularization. Various depth signals were explored to distill depth priors to the training of a NeRF [5, 6]. In the realm of 3DGS, depth regularization [23, 9] with global and local normalization is applied to constrain the 3D radiance field. SparseGS [10] uses depth priors and diffusion loss with a floater-pruning strategy to enhance the quality of renderings from unseen viewpoints. FSGS [11] grows Gaussians with a proximity-based Gaussian unpooling strategy regularized by depth. Previous approaches mainly target 3DGS training. SPARS3R [24]utilizes a pointmap estimator MASt3R [25] for 3DGS initialization, whereas we use monocular depths. Monocular estimates are typically sharper due to less constraints and have been shown to serve as a coarse solution for optimization in unposed reconstruction [26]. In addition, significant appearance variations pose challenges in MASt3R's view-consistent output. In our paper, we demonstrate the effectiveness of our proposed back-projected point cloud as an improved initialization strategy.

**Novel View Synthesis with varying appearances**    Casual photo collections often include images taken at different times or seasons, resulting in inconsistent appearances. To address the limitations of

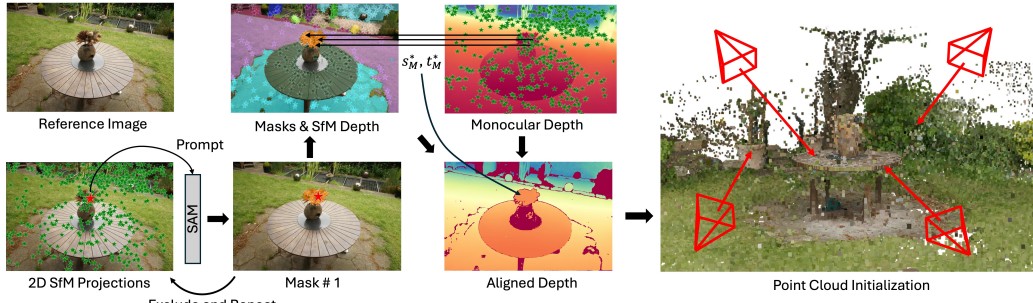

Figure 2: **Overview of our depth prior initialization of MS-GS.** Semantic masks and corresponding SfM point depth within each mask are obtained through our SfM-prompted Semantic module, detailed in Section 3. We then align monocular depth to SfM depth for each mask by computing the optimal scale $s_M^*$ and shift $t_M^*$. The point cloud is obtained from the back-projection of aligned depths and corresponding image pixel values to construct 3DGS initialization.

vanilla NeRF and 3DGS, which assume static scenes with consistent appearance, subsequent works integrate additional feature representations to account for appearance changes. NeRF in-the-wild (NeRF-W) [14] models static and transient volumes conditioned on image embeddings. Ha-NeRF [15] employs a convolutional neural network (CNN) to extract image appearance features and introduces a view-consistent loss to encourage consistent appearance across viewpoints. CR-NeRF [16] explores cross-ray features and their interaction with global image appearance for better appearance modeling. Recent adaptations of 3DGS for in-the-wild settings include GS-W [12], which uses an adaptive sampling strategy based on 2D feature maps to capture both dynamic and intrinsic appearance for each Gaussian. Wild-GS [13] extends feature representation to 3D as triplane features by incorporating rendered depth with input images. WildGaussians [17] leverages appearance embedding and DINO [27] features to handle appearance changes and occlusions with 3DGS. While these approaches are effective when abundant images are available, their performance severely degrades with sparse inputs, necessitating further advances in this line of research.

## 3 Method

Our proposed method, MS-GS, builds on the efficient 3DGS framework, in which a scene is represented by a set of Gaussian primitives $\{\mathcal{G}_i\}_{i=1}^N$. Each Gaussian $\mathcal{G}_i$ is defined with the learnable parameters: $xyz$ position $\mu_i \in \mathbb{R}^3$, opacity $\alpha_i \in \mathbb{R}$, color $c_i \in \mathbb{R}^3$, scale $s_i \in \mathbb{R}^3$, and quaternion $q_i \in \mathbb{R}^4$ for rotation. To improve its robustness in sparse-view synthesis and multi-appearance modeling, MS-GS consists of two parts: Semantic Depth Alignment first constructs a dense point cloud by expanding SfM points based on monocular depth and their semantic relevance (Section 3.1), illustrated in Fig. 2. MS-GS then introduces a series of geometry-guided supervisions based on 3D warping at a fine-grained pixel level and coarse feature level. (Section 3.2).

### 3.1 Semantic depth alignment

Gaussian-Splatting-based methods rely on discrete optimization to densify and prune Gaussians to fit the scene. Such discrete optimization is non-smooth and easily gets stuck at local minima; therefore, a good initial point cloud is crucial and provides anchors from which densification occurs. Conventionally, such a point cloud initialization comes from a prior SfM process and is assumed to be reasonably dense. In sparse-view or view-inconsistent scenarios, this assumption is often invalid due to insufficient correspondences. Therefore, we seek to densify the initial sparse point cloud based on monocular depth estimation.

**SfM-anchored alignment** After camera calibration, we have a set of $\mathcal{N}$ images $\{I_n | n = 1, 2, ..., \mathcal{N}\}$, an initial SfM point cloud $X \in \mathbb{R}^{P \times 3}$ and the camera poses. Applying world-to-camera transformation $W$ and camera intrinsics $K$ provides us with projected points $\mathcal{X} \in \mathbb{R}^{\bar{P} \times 2}$ with the pixel coordinate $u_n(i), v_n(i)$ and depth $d_n^{\text{sfm}}(i)$ of $x_i \in \mathcal{X}$ on image $I_n$. Note that from the

calibration process, we have a visibility function that indicates if $x_i$ is visible in $I_n$, preventing e.g. points from behind a wall from being projected.

Given a monocular depth estimation model [19], we can obtain a dense depth $D_n^{\text{mono}}$ for image $n$. While $D_n^{\text{mono}}$ is not in SfM scale and is not multi-view consistent, $d_n^{\text{sfm}}$ can be used to align $D_n^{\text{mono}}$. Specifically, a Least-Squares formulation is solved to find the optimal scale $s_n^*$ and shift $t_n^*$ for such alignment:

$$s_n^*, t_n^* = \arg \min_{s,t} \|s \cdot D_n^{\text{mono}}(u_n(i), v_n(i)) + t - d_n^{\text{sfm}}(i)\|_2 \, \forall i, \tag{1}$$

where $\| \cdot \|_2$ denotes the Euclidean norm. Once $s_n^*$ and $t_n^*$ are estimated, the aligned monocular depth $D_n^{\text{mono*}} = s_n^* \cdot D_n^{\text{mono}} + t_n^*$ can be back-projected into space to form a dense point cloud with all images by

$$X^{\text{mono}} = \bigcup_{n=1}^{\mathcal{N}} W^{-1} K^{-1} D_n^{\text{mono*}}(u, v) \, \forall u, v \in I_n. \tag{2}$$

While this formulation can be efficiently computed to construct a very dense point cloud, such a point cloud will be very noisy. Firstly, while the estimated $D_n^{\text{mono}}$ may be visually pleasing and detailed, the relative depth between objects within an image is often inaccurate due to its inherent ambiguity. Secondly, in a sparse-view scenario, the number of reliable SfM points is limited. Even if the error in Eq. (1) is minimized, it's unclear whether regions without sufficient constraints, i.e. $d_n^{\text{sfm}}$, are properly aligned. A noisy $X^{\text{mono}}$ does not improve NVS quality, as the dense but inaccurate points give rise to artifacts due to noisy gradients and lead to overfitting. To eliminate unreliable depth estimation in the alignment process, we propose an SfM-prompted Semantic Alignment scheme.

**SfM-prompted semantic alignment** We propose finding semantic regions enclosed by depth discontinuity using projected SfM points and performing individual alignment. As shown in Fig. 2, this is an iterative refinement process. Given a set of visible $\bar{P}$ SfM points $x_i \in \mathcal{X}$ projected on image $I_n$, we take a point $x_i$ and predict its semantically relevant region through an interactive segmentation model [21] $\mathcal{S}$, i.e. $M_i = \mathcal{S}(x_i, I_n)$. The intersection SfM points between $\mathcal{X}$ and $M_i$ are represented as $x_{m,i}$, which are semantically related to $x_i$ within the mask. We determine if $M_i$ has enough support by a threshold on $|x_{m,i}|$, the number of SfM points in the mask. To address the situations where insufficient points are caused by partial mask prediction, a second pass is performed $M_i = \mathcal{S}(x_{m,i}, I_n)$ if the threshold is not met the first time. Additionally, masks are checked for merging if they largely overlap to ensure semantic completeness. After each iteration, $x_{m,i}$ are removed from $\mathcal{X}$, and another $x_i$ is sampled until empty, and we obtain a set of final masks $\{M_{\text{final}}\} \in \mathbb{R}^{\mathcal{M} \times H \times W}$ in the end. The detailed algorithm is presented in the supplement.

Similarly, an optimal scale $s_m^*$ and shift $t_m^*$ are computed to align monocular depth $D_m^{\text{mono}}$ and SfM depth $d_m^{\text{SfM}}$ for each mask:

$$s_m^*, t_m^* = \arg \min_{s,t} \|s \cdot D_m^{\text{mono}}(u_m(i), v_m(i)) + t - d_m^{\text{sfm}}(i)\|_2 \, \forall i,$$
$$D_m^{\text{mono*}} = s_m^* \cdot D_m^{\text{mono}} + t_m^* \tag{3}$$

The point cloud is aggregated from all masks in each image through back-projection to initialize 3D Gaussians:

$$X^{\text{mono}} = \bigcup_{j=1}^{\mathcal{N}} \bigcup_{m=1}^{\mathcal{M}} W^{-1} K^{-1} D_m^{\text{mono*}}(u_m, v_m) \, \forall u_m, v_m \in M_{\text{final}}. \tag{4}$$

### 3.2 Multi-view geometry-guided supervisions

Modeling multi-appearance scenes under sparse-view constraints is especially difficult: view-specific lighting and weather variations demand more observations to disentangle appearance from structure. Consequently, overfitting, where the model memorizes the sparse training images instead of learning view-invariant geometry, becomes more severe than in the single appearance setting, as the SoTA methods for unconstrained settings exhibit obvious floaters and appearance inconsistencies. To encourage 3D consistency and appearance regularization, our method, illustrated in Fig. 3, exploits virtual cameras for multi-view supervision and utilizes 3D warping to establish correspondences for fine-grained pixel loss and coarse feature loss.

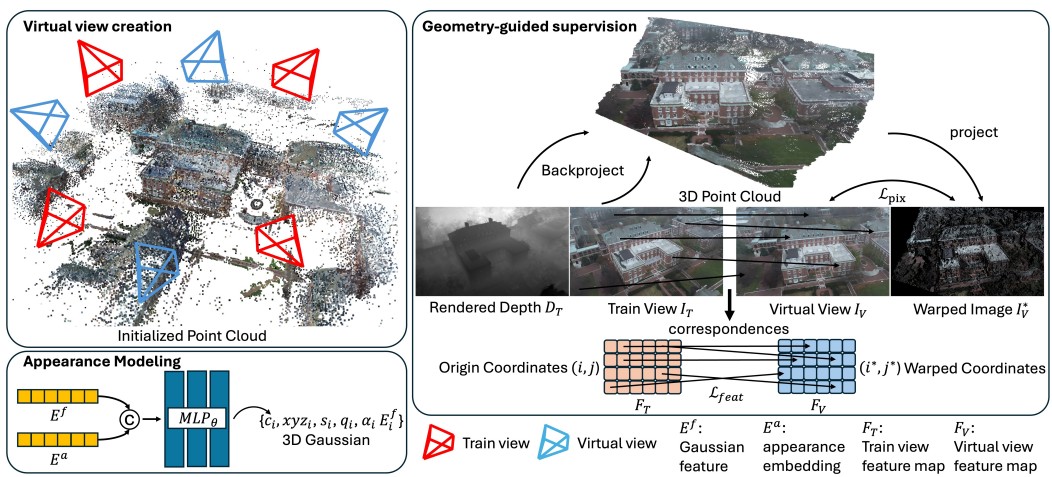

Figure 3: **Overview of our multi-view geometry-guided supervision of MS-GS.** Initialized from our proposed dense point cloud, we first create virtual views between training cameras. A 3D point cloud is back-projected given a training view $I_T$ and its corresponding rendered depth $D_T$, and then forward-projected onto the virtual view to obtain the warped image $I_V^*$ for a pixel loss. The correspondences from $I_T$ to $I_V^*$ are mapped to feature maps extracted from these two images to form a feature loss.

**Appearance modeling**   To handle appearance variations, MS-GS uses per-image appearance embedding $E^a \in \mathbb{R}^{\mathcal{N} \times 32}$ for optimization, where $\mathcal{N}$ is the number of images.

Meanwhile, MS-GS models a canonical scene representation through per-Gaussian feature embeddings $E^f \in \mathbb{R}^{N \times 16}$, where $N$ is the number of Gaussians. A feature fusion network $MLP_\theta$ takes these two appearance components to decode the RGB colors of 3D Gaussians $c \in \mathbb{R}^{N \times 3}$:

$$c = MLP_\theta \left( E^a, E^f \right). \tag{5}$$

**Virtual view creation**   The scene-reconstruction task is under-constrained when limited viewpoints are available. To introduce multi-view regularization, we create virtual views, which enable the additional supervision detailed in the following sections. At each iteration, we interpolate the current training view toward one of its top-k nearest-neighbour views. Camera translations are linearly interpolated with a weight in $[0, 1]$, while rotations are blended with SLERP [28]. The field of view (FOV) is also interpolated to avoid empty regions when moving between near-/far or wide-/narrow FOV pairs. Crucially, each virtual view uses the appearance embedding of the current training image, ensuring that it renders the same appearance and allows the optimization of the same embedding.

**Pixel warping supervision**   Leveraging the geometry optimized from our proposed dense initialization, we create supervision based on 3D warping [29, 30, 31]. We warp one view onto another with known depth and camera matrices. We first render the training and its virtual view to get colors $I_T, I_V$ and depths $D_T, D_V$. The training view pixels are back-projected to 3D points, then forward-projected onto the virtual view to produce a warped image $I_V^*$. Furthermore, pixels that have smaller values in the rendered depth $D_V$ than those in the warped depth are removed due to occlusion, forming a mask $M_{ocl}$. This explicit pixel-wise loss is formulated as:

$$\mathcal{L}_{\mathrm{pix}} = \| M_{\mathrm{ocl}} \odot (I_V - I_V^*) \|_1 . \tag{6}$$

Such a loss allows supervision from multiple views by leveraging reasonably accurate depth to constrain the geometry and appearance along the newly created rays at virtual cameras, as if a "floater" Gaussian exists that will alter the re-projected pixel color.

**Semantic feature supervision**   While the pixel loss provides fine-grained supervision at virtual views, blank pixels are produced due to occlusions and rounding errors when moving viewpoints. Thus, we propose to use a coarse semantic feature supervision at the local patch level, i.e, the receptive field of each feature-map element. Given a feature extractor [32], we can obtain the feature maps of

the training view $F_T$ and virtual view $F_V$. We make use of the geometry correspondences from 3D warping once again for more effective supervision. Formally, the feature map of the training view $F_T$ is transformed to $F_V^*$, which is computed using cosine distance loss with $F_V$:

$$\mathcal{L}_{\text{feat}} = \text{dist}\,(F_V, F_V^*)\,, \text{ where } F_V^*(i^*, j^*) = F_T(i, j), \tag{7}$$

where $(i, j)$ and $(i^*, j^*)$ are origin pixel coordinates on $I_T$ and warped pixel coordinates on $I_V$. The correspondences $(i, j) \rightarrow (i^*, j^*)$, illustrated in Fig. 3, are mapped to the feature map resolution from image resolution.

**Optimization**    Incorporating all the aforementioned techniques, the training objective of MS-GS is:

$$\mathcal{L}_{\text{total}} = \lambda_I \|I_T - I_T^*\|_1 + (1 - \lambda_I)\text{SSIM}(I_T, I_T^*) + \lambda_{\text{pix}}\mathcal{L}_{\text{pix}} + \lambda_{\text{feat}}\mathcal{L}_{\text{feat}}, \tag{8}$$

where $I_T^*$ is the ground-truth training image. Notably, the geometry-guided supervisions start after the scene converges to a sufficiently accurate geometry to establish correspondences.

## 4    Experiments

### 4.1    Datasets

We evaluate the performance of MS-GS and current SoTA methods on three real-world scenes with sparse inputs–one with single appearance and two with varying appearances. **Sparse Mip-NeRF 360 Dataset [33]** contains 4 outdoor and 4 indoor scenes with a complex central object or area and a detailed background. We sampled 20 images from each scene for training. **Sparse Phototourism Dataset [22]** consists of scenes of well-known monuments. Specifically, we use "Brandenburg Gate", "Sacre Coeur", and "Trevi Fountain", following previous works [14, 12, 13, 17]. We sampled 20 images from the official training set and kept the same testing split for evaluation. Note that these are 2.62%, 2.41%, and 1.18% of the full training set for each scene, respectively. **Sparse Unbounded Drone Dataset**: We collected drone footage of 3 different buildings captured in orbit, creating a dataset featured in multi-view and multi-appearance scenarios. The dataset includes 4 distinct appearances: sunny, cloudy, snowy, and low-light, each captured with a full 360° view. We evenly sampled 5 images from each appearance, resulting in 20 images for training each scene. We aim to establish these benchmarks for sparse-view synthesis in unconstrained settings. Please find a fuller description of our dataset in the supplement.

### 4.2    Evaluation and Implementation

Most sparse-view synthesis methods [34, 3, 4, 6, 7, 8, 5, 9] assume ground-truth (GT) camera poses, i.e., calibration with dense views, thereby bypassing the challenges of registration and point triangulation. Thus, we propose an in-the-wild evaluation setting to evaluate sparse-view synthesis. A coordinate alignment method, provided in the supplement, is developed to perform separate registrations that disentangle training and testing images and then align them in the same coordinate system. Note that a slight pixel offset occurs during pose alignment, which disturbs pixel-based metrics PSNR and SSIM [35]. Therefore, we evaluate only with perceptual metrics LPIPS [36] and DreamSim (DSIM) [37] in this setting. We strongly encourage the readers to inspect the supplement for more details and analysis on the in-the-wild evaluation, metrics, and implementation.

Table 1: Ablation studies on different components of MS-GS. The metrics are reported as the average on the Sparse Unbounded drone dataset; **bold** numbers are the best, underscored second best.

| Dense Init. | $\mathcal{L}_{\text{pix}}$ | $\mathcal{L}_{\text{feat}}$ | PSNR↑ | SSIM↑ | LPIPS↓ | DSIM↓ |
|:---:|:---:|:---:|:---:|:---:|:---:|:---:|
| ✗ | ✗ | ✗ | 18.49 | 0.492 | 0.367 | 0.151 |
| ✓ | ✗ | ✗ | 19.29 | 0.538 | 0.336 | 0.115 |
| ✓ | ✓ | ✗ | 19.63 | 0.564 | 0.330 | 0.104 |
| ✓ | ✗ | ✓ | 19.65 | 0.569 | 0.328 | 0.104 |
| ✓ | ✓ | ✓ | **19.87** | **0.580** | **0.322** | **0.096** |

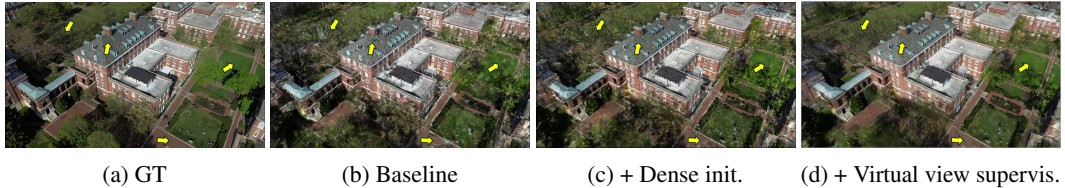

| (a) GT | (b) Baseline | (c) + Dense init. | (d) + Virtual view supervis. |

Figure 4: Novel view synthesis results when components are added sequentially. Please zoom in if possible for better visualization.

## 4.3 Ablation Study

We conduct an ablation study to validate the effectiveness of our method in Table 1 and Fig. 4. We refer to 3DGS augmented with multi-appearance capabilities using per-image embeddings and Gaussian feature embeddings as the baseline and report its metrics in the first row of Table 1. We identify that incorporating our semantic depth alignment initialization significantly improved the metrics with 0.8 dB in PSNR, 0.046 in SSIM, -0.031 in LPIPS, and -0.036 in DSIM. This result proves our hypothesis that optimizing 3DGS directly on a sparse SfM point cloud leaves ambiguities: novel views display incomplete surfaces and artifact Gaussians, shown in Fig. 4b. By contrast, our semantically dense initialization widens the solution space and provides reliable geometric support, enabling more effective Gaussian densification and pruning. This approach regularizes the scene structure and yields a more complete geometry, e.g., filling in the holes on the rooftop.

Next, adding 3D warping pixel loss and geometry-guided feature loss based on virtual views each further boosts the performance by a similar margin. The complete multi-view supervisions enhance the metrics by 0.58 dB in PSNR, 0.042 in SSIM, -0.014 in LPIPS, and -0.019 in DSIM. As visualized in Fig 4d, this strategy suppresses residual artifacts, on regions such as grass, trees, and road markings, and renders a more faithful reconstruction. The multi-view supervision enforces radiance consistency and also refines geometry through synergistic feedback. All proposed components are complementary, and the best results are achieved when combined. The analysis and visualization for semantic scaling validation can be found in the appendix.

## 4.4 Comparisons

Table 2: Quantitative Comparison on sparse Mip-NeRF 360 dataset; **bold** numbers are the best, underscored second best. We only evaluate LPIPS and DSIM for in-the-wild setting as discussed in the evaluation section and supplement.

|  | In the Wild | | GT Pose | | | |
|---|---|---|---|---|---|---|
| Method | LPIPS↓ | DSIM↓ | PSNR↑ | SSIM↑ | LPIPS↓ | DSIM↓ |
| DRGS[23] | 0.588 | 0.273 | 19.16 | 0.516 | 0.544 | 0.253 |
| DNGS[9] | 0.503 | 0.193 | 19.79 | 0.588 | 0.466 | 0.185 |
| SparseGS[10] | 0.309 | 0.105 | 21.37 | 0.667 | 0.260 | 0.093 |
| FSGS[11] | 0.327 | 0.098 | 21.67 | 0.637 | 0.394 | 0.128 |
| SPARS3R[24] | 0.245 | 0.082 | 21.48 | 0.674 | 0.213 | 0.081 |
| **Ours** | **0.238** | **0.080** | **22.39** | **0.702** | **0.211** | **0.072** |

**Sparse Mip-NeRF 360 Dataset** As shown in Table 2 and Fig 5, our approach demonstrates notable improvements over other 3DGS-based methods. DRGS and DNGS yield overly smooth renderings because of the suboptimal depth regularization. Although SparseGS and FSGS improve the rendering quality through floater pruning, score distillation regularization, and the densification strategy. Although SPARS3R renders more details, the global alignment can leave regions where points are incorrectly placed, manifested as artifacts due to the strong initialization bias. In comparison, MS-GS favors more accurate local regions, and our virtual view supervision further improves the results. As in Fig. 5, it preserves coherent geometry and reconstructs fine details, such as the table legs and carpet. These results show the efficacy of our semantic dense initialization in regularizing scene structure and facilitating the optimizations of the 3DGS framework.

Table 3: Quantitative Comparison on sparse unbounded drone dataset. Methods[†] renders each test view with the appearance embedding taken from the training image that is nearest in pose and shares the same appearance. Please refer to the setting described in Sec. A.1.2. Other methods extract appearance from the input image.

| Method | Computation | | In the Wild | | GT Pose | | | |
|---|---|---|---|---|---|---|---|---|
| | GPU hrs. | FPS | LPIPS↓ | DSIM↓ | PSNR↑ | SSIM↑ | LPIPS↓ | DSIM↓ |
| NeRF-W[†][14] | 5.79 | <1 | 0.659 | 0.451 | 16.95 | 0.453 | 0.621 | 0.402 |
| Ha-NeRF [15] | 11..66 | <1 | 0.669 | 0.405 | 16.27 | 0470 | 0.622 | 0.361 |
| CR-NeRF [16] | 7.31 | <1 | 0.694 | 0.489 | 16.59 | 0.467 | 0.612 | 0.370 |
| GS-W [12] | 2.75 | 63 | _0.446_ | _0.213_ | _17.33_ | _0.491_ | _0.487_ | _0.279_ |
| Wild-GS [13] | _1.01_ | 74 | 0.526 | 0.425 | 14.13 | 0.345 | 0.547 | 0.487 |
| WildGaussians[†][17] | 2.57 | _170_ | 0.502 | 0.302 | 15.60 | 0.388 | 0.546 | 0.428 |
| **Ours[†]** | **0.29** | **373** | **0.331** | **0.105** | **19.87** | **0.580** | **0.322** | **0.096** |

Table 4: Quantitative Comparison on sparse Phototourism dataset. Methods[‡] optimize appearance embedding on the left half of the test image and evaluate on the other half. Other methods extract appearance from the input image.

| Method | Computation | | In the Wild | | GT Pose | | | |
|---|---|---|---|---|---|---|---|---|
| | GPU hrs. | FPS | LPIPS↓ | DSIM↓ | PSNR↑ | SSIM↑ | LPIPS↓ | DSIM↓ |
| NeRF-W[‡][14] | 8.28 | <1 | 0.325 | 0.180 | _17.93_ | 0.619 | 0.439 | 0.269 |
| Ha-NeRF [15] | 14.49 | <1 | 0.310 | 0.167 | 16.33 | 0.663 | 0.446 | _0.231_ |
| CR-NeRF [16] | 8.83 | <1 | 0.299 | _0.145_ | 16.98 | _0.668_ | 0.422 | 0.232 |
| GS-W [12] | 2.32 | 57 | _0.289_ | 0.161 | 16.74 | 0.637 | _0.365_ | 0.245 |
| Wild-GS [13] | _0.95_ | 70 | 0.331 | 0.216 | 16.21 | 0.592 | 0.385 | 0.330 |
| WildGaussians[‡][17] | 2.71 | _162_ | 0.317 | 0.195 | 14.33 | 0.577 | 0.433 | 0.248 |
| **Ours[‡]** | **0.32** | **351** | **0.258** | **0.138** | **18.99** | **0.684** | **0.269** | **0.154** |

**Sparse unbounded drone and Phototourism Datasets**    Tables 3 and 4 together with Fig. 5 present results on these benchmarks. On the sparse unbounded-drone dataset, our approach significantly outperforms the SoTA methods with improvements of 2.54 dB in PSNR, 0.089 in SSIM, and cuts LPIPS and DSIM by 33.8% and 65.6%, respectively, with respect to the best prior method. The added capacity of U-Net in GS-W and Wild-GS is hard to train with sparse views and worsens the overfitting issue, manifesting as artifacts and blurred structure, e.g., the building surfaces. Without sufficient constraints, the appearance-affine head and uncertainty weighting in WildGaussians can absorb photometric error instead of correcting structures, leaving as off-view aliasing and texture drift. All these methods exhibit inconsistent appearances and floaters, e.g., the statue and the dome in Trevi Fountain and Brandenburg Gate. In contrast, MS-GS reconstructs a more coherent structure with fine-grained details and consistent appearance rendering, thanks to the synergy of our proposed components. Furthermore, our design is lightweight, requiring >3× less GPU time for training over Wild-GS and rendering at 300+ FPS.

## 5    Limitations

First, MS-GS is not designed for handling transient objects, which is especially difficult under sparse views due to increased uncertainty and ambiguities in scene reconstruction. While recent methods leverage uncertainty masks to remove transients and allow other observations to fill in the blank, often no other observations exist under a sparse setting; therefore, the transient regions remain under-constrained. Insufficient learning of transient masks often leads to worse results. Second, generalizing to non-Lambertian surfaces is challenging and requires more complex modeling. As MS-GS targets 3D consistency between views, the specular highlights can be smoothed or averaged out (see Fig. 10). Combining our framework with techniques, such as explicitly modeling of light [38] and surface reconstruction [39], remains an open research area. Specific techniques have to be developed to solve these limitations, which we leave as future work.

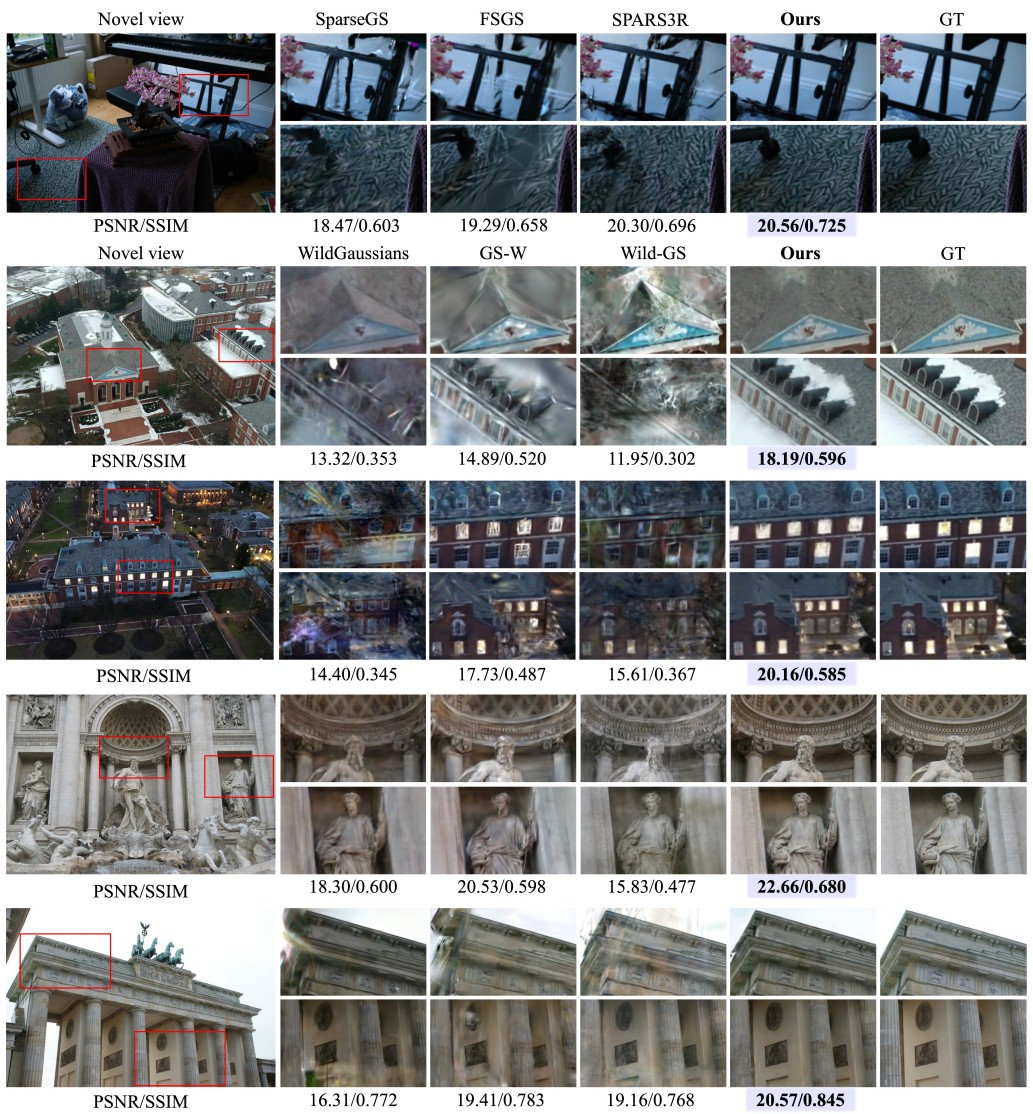

Figure 5: Qualitative comparison of novel view synthesis across different datasets. MS-GS (ours) excels at capturing detailed structures and preserving consistent appearance.

# 6 Conclusion

MS-GS establishes a strong baseline for multi-appearance sparse-view 3D Gaussian Splatting, significantly improving over existing methods. We identify that one of the limitations of 3DGS-based methods in sparse-view synthesis is the sparse point cloud initialization. To address this, our proposed method constructs a dense point cloud by performing individual alignment and back-projection in local semantic regions. The geometric prior steers the 3DGS optimization and acts as the cornerstone for our multi-view geometry-guided supervision. We further introduce virtual views to provide supervision along newly created camera rays as self-regularization to suppress floaters and encourage consistency, which aligns with the fundamental constraint of 3D reconstruction. Jointly, MS-GS offers a robust solution under challenges of limited viewpoints and varying appearances that naturally arise in real-world data.

# 7 Acknowledgement

This research is based upon work supported by the Office of the Director of National Intelligence (ODNI), Intelligence Advanced Research Projects Activity (IARPA), via IARPA R&D Contract No. 140D0423C0076. The views and conclusions contained herein are those of the authors and should not be interpreted as necessarily representing the official policies or endorsements, either expressed or implied, of the ODNI, IARPA, or the U.S. Government. The U.S. Government is authorized to reproduce and distribute reprints for Governmental purposes notwithstanding any copyright annotation thereon. This research and/or curriculum was supported by grants from NVIDIA and utilized NVIDIA RTX A5500 and A100 GPUs.

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

# A  Technical Appendices and Supplementary Material

## A.1  Sparse unbounded drone dataset

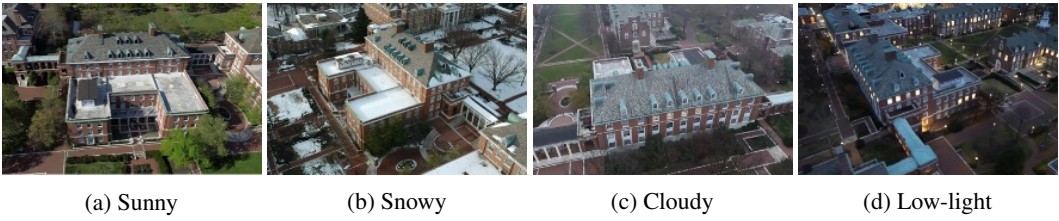

| (a) Sunny | (b) Snowy | (c) Cloudy | (d) Low-light |

Figure 6: Dataset visualizations

### A.1.1  Overview and registration

As shown in Fig. 6, our dataset consists of 4 unique appearances: sunny, snowy, cloudy, and low-light, captured in drone footage from different seasons and time of the day. We sampled 5 images per appearance evenly from the circular camera trajectory, resulting in 20 training images, and similarly for 12 testing images. We used hloc [40] registration pipeline with SuperPoint features [41] and RoMa [42] matcher.

### A.1.2  Comparison with prior benchmark

To obtain the appearance embeddings of a test image from the Phototourism [22] dataset, prior setups either optimize on half of the test image or use a network to extract features from the entire test image. Both of these approaches involve the use of test images during evaluation. For our dataset, the multi-view, multi-appearance setup enables the correct appearance to be rendered based on metadata/timestamp, not pixel-wise information from the test image. For example, a snowy test image can be rendered with the appearance embedding of a snowy training image by relating their timestamps. This setup is also faster than optimizing half of the test image. Additionally, our dataset contains scenes with 360-degree coverage by perspective cameras, whereas Phototourism is covered by face-forward images.

## A.2  Experiments and visualizations

### A.2.1  Initialization comparisons

Table 5: Experiments on the effectiveness of different initializations. The metrics are reported as the average on the Sparse Mip-NeRF 360 dataset.

| Method | PSNR↑ | SSIM↑ | LPIPS↓ | DSIM↓ |
|---|---|---|---|---|
| sparse init. | 20.83 | 0.627 | 0.267 | 0.109 |
| image-level alignment | 21.06 | 0.631 | 0.253 | 0.094 |
| semantic alignment | **21.96** | **0.690** | **0.216** | **0.080** |
| DUSt3R [43] | 19.89 | 0.585 | 0.270 | 0.118 |

In this section, we compare different initialization strategies, namely the sparse SfM point cloud, dense initialization with image-level alignment as introduced in Section 3.1, our proposed dense initialization with semantic alignment, and DUSt3R [43] point cloud. In Figure 7, the sparse SfM point cloud contains only a few thousand points and covers a small fraction of the scene. Even though initialization with image-level alignment is much denser, it also introduces more errors in the point cloud, leading to noisy structures. In contrast, our method, which favors more accurate local alignments, achieves cleaner and semantically meaningful scene components.

Quantitatively (Table 5), initialization from image-level alignment offers only marginal benefit compared to the baseline, as misplaced Gaussians that are not pruned or densified correctly can produce noisy structures, as shown in Fig. 8. DUST3R is a two-view pointmap estimator. When the number of images is greater than two, it aggregates all pairwise pointmap predictions into a very

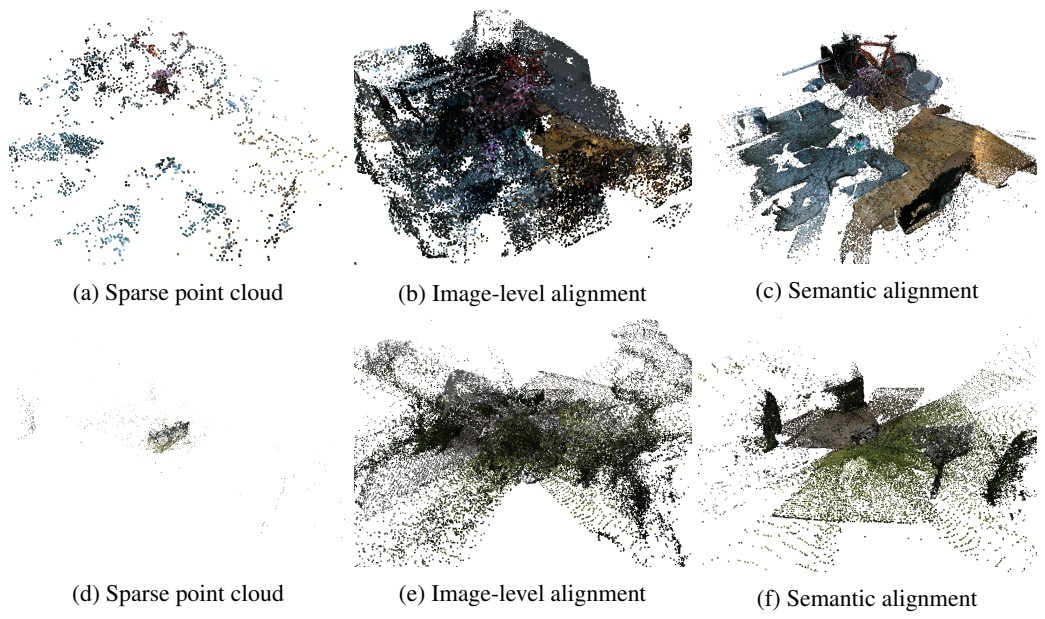

(a) Sparse point cloud      (b) Image-level alignment      (c) Semantic alignment

(d) Sparse point cloud      (e) Image-level alignment      (f) Semantic alignment

Figure 7: Visualizations of different point cloud initializations.

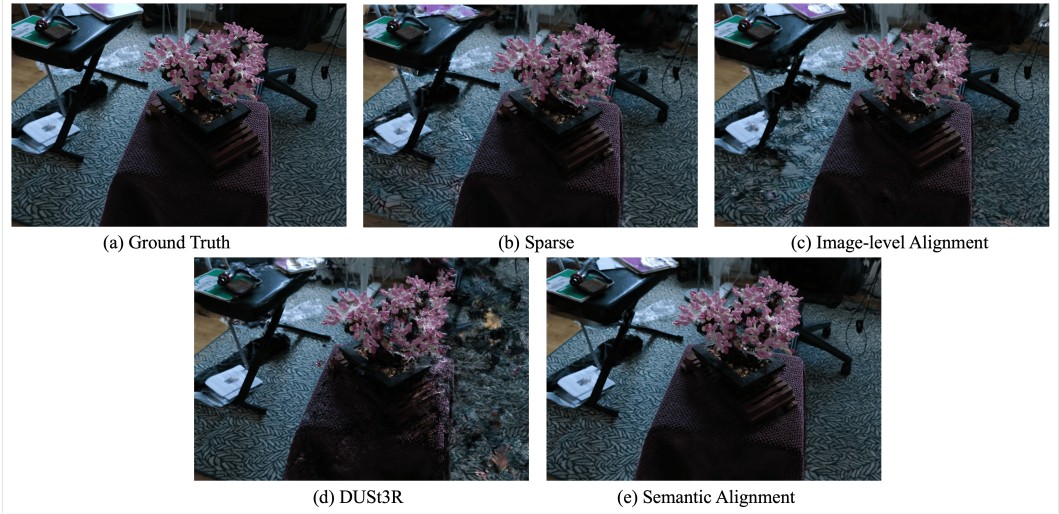

(a) Ground Truth      (b) Sparse      (c) Image-level Alignment

(d) DUSt3R      (e) Semantic Alignment

Figure 8: Visualizations of rendering with different point cloud initializations.

dense point cloud, usually millions of points. To utilize DUSt3R points, we align them to the SfM points based on corresponding pixels using Procrustes Alignment [44]. While the output of DUSt3R is visually pleasing, it still suffers from depth ambiguities, leading to incorrect placement of objects. As shown in Fig. 8, it produces ghosting artifacts due to the strong initialization bias. Notably, our approach improves the PSNR by 1.13, SSIM by 0.063, LPIPS by 0.051, and DSIM by 0.029. In addition, the time complexity of DUSt3R to run N images is $\mathcal{O}(N^2)$ compared to $\mathcal{O}(N)$ for the monocular depth estimator, which makes it harder to scale. This analysis highlights the importance of semantic depth alignment, which guides 3DGS to converge to a better scene reconstruction.

We also validate the semantic alignment accuracy on the bonsai scene. Specifically, we use the Multi-View Stereo (MVS) depth from dense views as GT depth estimates. An image-level estimate of scale results in a mean absolute error of 1.94, and our per-semantic-region scaling reduces this to 1.07. The error maps are visualized in Fig. 9, which shows that the depth continuities around object boundaries exhibit higher error, and the error inside objects is lower/darker with our piece-wise alignment approach.

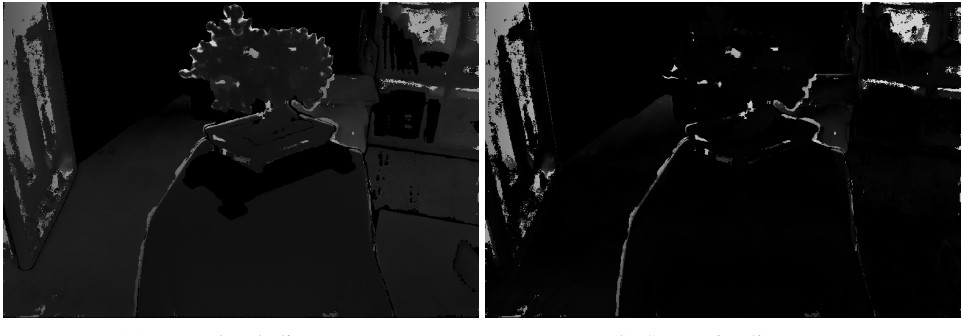

(a) Image-level alignment        (b) Semantic alignment

Figure 9: Error maps after alignment. Brighter means higher error.

### A.2.2 Edge cases

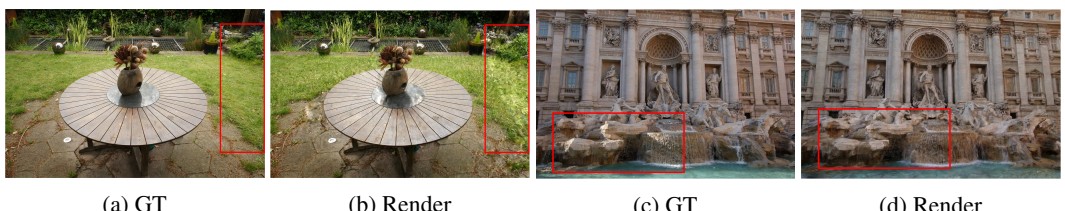

(a) GT      (b) Render      (c) GT      (d) Render

Figure 10: As MS-GS favors more accurate local alignment, areas without dense initialization can introduce artifacts in (a) and (b). Specular highlights can be smoothed out due to the multi-view consistency and limited capacity of appearance embedding, as seen in (c) and (d).

### A.3 Implementation details

We develop MS-GS based on the 3DGS implementation from NeRFStudio, called Splatfacto [45]. The baseline introduced in our ablation study Section 4.3 uses the same Splatfacto model. In Semantic Depth Alignment, the minimum number of SfM points threshold within a valid mask is 10. The intersection of two masks for merging is 0.7. We use both back-projected point cloud and MVS points for our initialization. The appearance MLP consists of 3 layers of 64 hidden units. The embedding sizes for the Gaussian feature and per-image appearance embeddings are 16 and 32, respectively. Virtual views are generated by interpolating toward one of the $k=4$ nearest training cameras. We use features extracted from blocks 3 and 4 of VGG-16 [32, 46, 47] for feature loss at different resolutions and receptive fields. We set $\lambda_I = 0.8$, $\lambda_{\text{pix}} = 1.0$, and $\lambda_{\text{feat}} = 0.04$. The total number of training iterations is 16,500, with the geometry-guided supervision enabled after 15,000 iterations. The same hyperparameters are maintained throughout the experiments. Results are obtained with the NVIDIA RTX A5500 GPU.

### A.4 Appearance embedding initialization

Table 6: Experiments on the appearance embedding initializations. The metrics are reported as the average on the Sparse Unbounded Drone dataset.

| Method | PSNR↑ | SSIM↑ | LPIPS↓ | DSIM↓ |
|---|---|---|---|---|
| normal distribution in [0,1] | 18.77 | 0.524 | 0.352 | 0.132 |
| near-zero initialization | **19.29** | **0.538** | **0.336** | **0.115** |

Appearance embeddings are typically initialized with a normal distribution in $\begin{bmatrix} 0, 1 \end{bmatrix}$ [48, 14]. We find that this initialization introduces view-specific biases. Instead, we initialize them near zero, i.e., uniform distribution in $\begin{bmatrix} -1 \times 10^{-4}, 1 \times 10^{-4} \end{bmatrix}$, which shows improved metrics and yields meaningful clusters after training, as shown in Table 6 and Fig. 11. We attribute this result to the near-zero

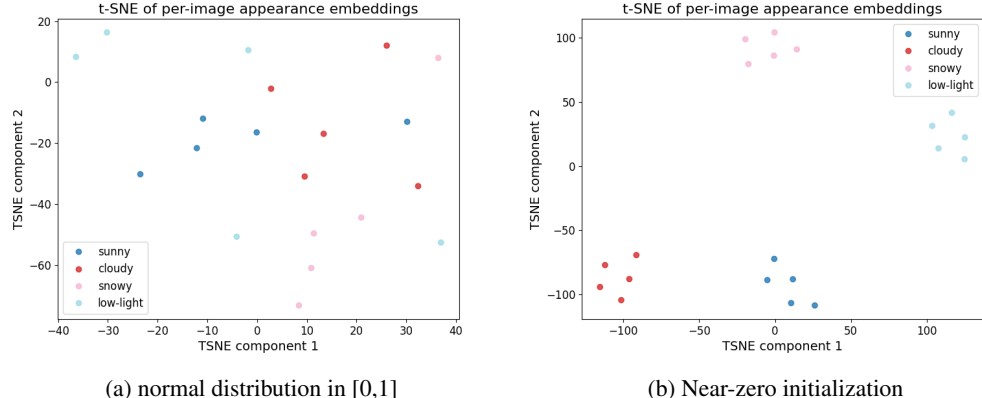

(a) normal distribution in [0,1]     (b) Near-zero initialization

Figure 11: t-SNE visualizations of per-image appearance embeddings after training with different initializations

initialization: it delays the expressive power of the per-image appearance embeddings, minimally influencing the MLP training in the early stages, so the network first learns a shared color basis and later allocates capacity to disentangle appearances.

### A.4.1 Sparser setting

Table 7: Experiments in 12-view setting, where each appearance has 3 images. MS-GS continues to outperform other methods. The metrics are reported as the average on the Sparse unbounded drone dataset.

| Method | PSNR↑ | SSIM↑ | LPIPS↓ | DSIM↓ |
|---|---|---|---|---|
| GS-W [12] | 14.83 | 0.371 | 0.560 | 0.457 |
| Wild-GS [13] | 13.66 | 0.289 | 0.587 | 0.583 |
| WildGaussians [17] | 12.47 | 0.278 | 0.612 | 0.664 |
| Ours | 17.78 | 0.477 | 0.412 | 0.180 |

### A.4.2 Dense init. for other in-the-wild methods

In this section, we investigate the performance of other in-the-wild methods using our proposed dense initialization. Based on Table 8, on one hand, all methods achieve significantly better metrics compared to their original baseline. For example, GS-W gains 0.9 dB in PSNR, 0.054 in SSIM, and reduces 0.11 in LPIPS and 0.069 in DSIM. This experiment confirms that our initialization is a drop-in enhancement for 3DGS-based pipeline. On the other hand, the improved performance of other methods is still inferior to MS-GS by a large margin, validating the effectiveness of our appearance modules and multi-view geometry-guided supervision in this challenging setting.

Table 8: Experiments on the effectiveness of our dense initialization applied to other methods for multi-appearance synthesis. The metrics are reported as the average on the Sparse unbounded drone dataset.

| Method | PSNR↑ | SSIM↑ | LPIPS↓ | DSIM↓ |
|---|---|---|---|---|
| GS-W [12] | 17.33 | 0.491 | 0.487 | 0.279 |
| + dense init. | **18.23** | **0.545** | **0.377** | **0.210** |
| Wild-GS [13] | 14.13 | 0.345 | 0.547 | 0.487 |
| + dense init. | **14.35** | **0.395** | **0.544** | **0.443** |
| WildGaussians [17] | 15.60 | 0.388 | 0.546 | 0.428 |
| + dense init. | **16.50** | **0.449** | **0.482** | **0.316** |
| Ours | 19.87 | 0.580 | 0.322 | 0.096 |

### A.5 Semantic alignment algorithm

---

**Algorithm 1:** Semantic Masks Prediction

---

**Input** : Image $I_n$, a set of visible 2D SfM points $\mathcal{X}$ on $I_n$, segmentation model $\mathcal{S}$, threshold $\text{TH}_{\text{sfm}}$, threshold $\text{TH}_{\text{IoU}}$.

**Output** : Final set of masks $M_{\text{final}}$.

1 **Def** append_mask($M_i$, $M_{final}$, $TH_{IoU}$)**:**
2    merged = False;
3    **for** $M \in M_{final}$ **do**
4      **if** $M_i \cap M > TH_{IoU}$ **then**
5        $M = M \cup M_i$ ;                    /* Merge the masks */
6        merged = True;
7        break;
8      **end**
9    **end**
10    **if** *not merged* **then**
11      $M_i \rightarrow M_{\text{final}}$ ;               /* Append the mask to set */
12    **end**
13 $M_{\text{final}} = \varnothing$;
14 **while** $\mathcal{X}$ *is not empty* **do**
15    $x_i \sim \mathcal{X}$ ;                     /* Sample a point */
16    $M_i = \mathcal{S}(x_i, I_n)$ ;                /* Prompt a mask */
17    $x_{m,i} = \mathcal{X} \cap M_i$ ;           /* Find points within the mask */
18    **if** $|x_{m,i}| > TH_{sfm}$ **then**
19      append_mask($M_i$, $M_{final}$, $TH_{IoU}$) ;          /* Enough points */
20    **else**
21      $M_i = \mathcal{S}(x_{m,i}, I_n)$ ;       /* Re-prompt with points within the mask */
22      $x_{m,i} = \mathcal{X} \cap M_i$;
23      **if** $|x_{m,i}| > TH_{sfm}$ **then**
24        append_mask($M_i$, $M_{final}$, $TH_{IoU}$);
25      **else**
26        continue;
27      **end**
28    **end**
29    Exclude $x_{m,i}$ from $\mathcal{X}$ ;          /* Remove points from set */
30 **end**

---

The iterative refinement algorithm is detailed in Algorithm 1. This is an automatic process to find the semantic masks anchored by SfM points, which are back-projected individually to form a dense point cloud for 3DGS initialization.

### A.6 In-the-wild evaluation

Sparse-view and multi-appearance registration is challenging because of limited overlap and view inconsistency; Fewer reliable feature matches result in suboptimal pose estimation and point triangulation. Sparse-view methods [34, 3, 4, 6, 7, 8, 5, 9] commonly assume ground-truth camera poses, i.e., calibration from dense views. However, only training views should be available in an in-the-wild setting. 3DGS-based methods rely heavily on the SfM point cloud, further necessitating the separation of training and testing views from the registration stage. A previous approach [11] has tried to perform re-triangulation based on known train poses, but does not account for pose inaccuracy. Therefore, we propose a coordinate alignment method, illustrated in Fig. 12, to disentangle training and testing images in registration.

#### A.6.1 Coordinate alignment

In coordinate alignment, we seek to register training and testing views separately and align them together. Therefore, we perform two registrations: 1. training images only, resulting in the input

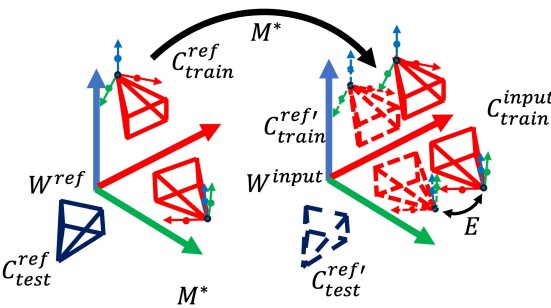

Figure 12: Illustration of Coordinate Alignment. We first compute the transformation $M^*$ between train cameras in two coordinate systems $C_{\text{train}}^{\text{ref}}$ and $C_{\text{train}}^{\text{input}}$; each camera corresponds to 4 points: one position and three rotation points, displayed as small black, red, green, and blue points in the figure. The transformed $C_{\text{train}}^{\text{ref}}$ is denoted as $C_{\text{train}}^{\text{ref'}}$ in dashed lines, which is used to compute the error $E$ between $C_{\text{train}}^{\text{input}}$. Finally, $M^*$ transforms test camera poses $C_{\text{test}}^{\text{ref}}$ to $C_{\text{test}}^{\text{ref'}}$ in the input coordinate system.

coordinate system $C_{\text{train}}^{\text{input}}$ 2. training and testing images in the reference coordinate system, $C_{\text{train}}^{\text{ref}}$ and $C_{\text{test}}^{\text{ref}}$, as SfM reconstructs the scene in a different coordinate system each time. A transformation $M^*$ is computed between $C_{\text{train}}^{\text{input}}$ and $C_{\text{train}}^{\text{ref}}$ using Procrustes Alignment [44] to transform test cameras $C_{\text{test}}^{\text{ref}}$ to the input coordinate system $C_{\text{test}}^{\text{ref'}}$. Conventionally, only camera positions/centers are considered during alignment. To leverage rotation information, we additionally sample three points along the camera's local rotation axes to form a small local frame around each camera center. Formally, the point set of each camera, represented as $P_{\text{cam}} \in \mathbb{R}^{4\times3}$, is defined as:

$$
R = [\, r_x, \, r_y, \, r_z \,] \in \mathbb{R}^{3\times3},
$$

$$
s = \sqrt{\sigma_x^2 + \sigma_y^2 + \sigma_z^2},
$$

$$
P_{\text{cam}} = \begin{bmatrix} T^\top \\ (T + s\,r_x)^\top \\ (T + s\,r_y)^\top \\ (T + s\,r_z)^\top \end{bmatrix} \in \mathbb{R}^{4\times3},
$$

(9)

where $R$ is the camera rotation matrix, $T \in \mathbb{R}^3$ is the translation (camera center), and $s$ is a scalar scaling factor approximated by the per-dimension standard deviation $\sigma$. Each row of $P_{\text{cam}}$ corresponds to a 3D point: the camera center followed by three axis-offset points derived from the camera's orientation. Finally, we use $C_{\text{train}}^{\text{input}}$, $C_{\text{test}}^{\text{ref'}}$, and points triangulated from $C_{\text{train}}^{\text{input}}$ for 3DGS input. In this way, we simulate the real-world scenario, where camera poses and 3D points are estimated from training views while having the testing camera poses in the same coordinate system for evaluation.

As shown in Table 9, our rotation-aware alignment reduces the rotation error $E_R$, in degrees, by more than 10 times and the position error $E_T$, in an arbitrary unit as in the SfM, by 4 times. This improvement results in accurately aligned test cameras and, consequently, more reliable evaluations.

Table 9: Experiments on the effectiveness of our rotation-aware camera alignment. The metrics are reported as the average on the Sparse Unbounded Drone dataset.

| Method | $E_R(\text{med})$ | $E_R(\mu)$ | $E_T(\text{med})$ | $E_T(\mu)$ |
|---|---|---|---|---|
| w/o rotation points | 0.791 | 0.793 | 0.0397 | 0.0377 |
| ours | **0.063** | **0.066** | **0.0067** | **0.0085** |

### A.6.2  Evaluation metrics

We evaluate the novel view rendering quality based on the image and perceptual metrics, including PSNR, SSIM [35], and LPIPS [36]. We also propose to use DreamSim (DSIM) [37] as an additional

metric, which is an ensemble method of different perceptual metrics [36, 49, 50, 51, 52, 53] fine-tuned for human visual perspectives.

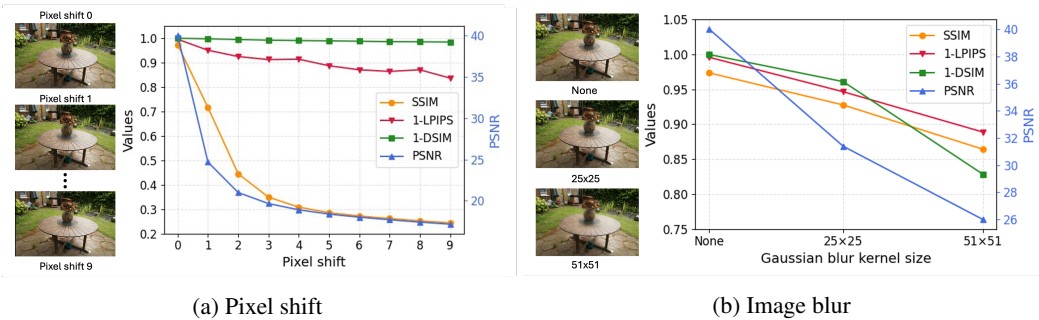

(a) Pixel shift

(b) Image blur

Figure 13: Evaluation of DSIM as metric

Our coordinate alignment method is accurate but not perfect, leaving small residual pose shifts. However, this slight pixel offset should not reflect a significant difference in metrics, dominating the quality assessment. As Fig. 13a shows, PSNR and SSIM drop steeply with a few pixel offsets, whereas DSIM remains almost flat. When images are dissimilar, where we add a blob of Gaussian blur at different kernel sizes in Fig. 13b to simulate semi-transparent Gaussians, DSIM shows a consistent decline as other metrics. This analysis indicates that DSIM is an appropriate metric for in-the-wild evaluations: it avoids over-penalising inevitable alignment errors while still capturing real perceptual degradation.

