# OpenReview forum: "MS-GS: Multi-Appearance Sparse-View 3D Gaussian Splatting in the Wild"
_NeurIPS.cc/2025/Conference — NeurIPS 2025 poster_

### Official Review · Reviewer_B6Ss · 2025-07-01

**Clarity:** 3
**Significance:** 3
**Originality:** 3
**Rating:** 4
**Confidence:** 4

**Summary:**

The paper proposes a collection of improvements to the standard Gaussian Splatting pipeline, particularly when the number of available input frames is small. The work focuses on (i) producing denser initialisation point clouds and (ii) generating and using virtual views as regularisers within the splat optimisation.

**Questions:**

My main questions / requests focus on the semantic scaling. I do not see why one would necessarily expect different regions of a single view depth output image to have different scaling factors, or for these scaling factors to necessarily be semantically-coherent. It would be great if the authors added some validation on this. Could could be done relatively simply -- take some images for which they have GT depth estimates and demonstrate that a single per-frame estimate of scale is less accurate than per-semantic-region estimates.

While I very much like the virtual view regularisation, it is not entirely obvious to me how reprojecting pixels from one real view into a virtual view, and then asking the splat to re-generate this (via the splat optimisation) is not already enforced by the fact that the splat optimisation aimed to generate those very same pixels for the real view.

Overall, given the good overall results and the two improvements proposed by the paper, I lean towards acceptance.

**Ethical Concerns:**

["NO or VERY MINOR ethics concerns only"]

**Final Justification:**

Following the rebuttal and the other reviews, I maintain my overall positive view of the paper and believe it should be accepted.

**Limitations:**

The authors do not appear to discuss the limitations and impact. The submission pdf appears to say that this is discussed in "section 6", yet I can only find sections up to 5.

**Quality:**

3

**Strengths And Weaknesses:**

Positive:
- The paper is well written and easy to understand.
- The approach improves results consistently in all comparisons included by the authors.
- The approach is quite quick, or at least significantly quicker than competing approaches, like Wild-GS.
- I found the virtual view idea to be quite novel and interesting.

Negative:
- While many of the results demonstrate clear improvements, in some other tables the numbers are quite close. An example is Table 1 (ie the ablation section of the results). It would be useful if the authors included variance in these cases.
- I could not find an ablation for the semantic scaling (i.e. as opposed for a single scalar for entire depth images).
- Also related to the semantic scaling, I am not convinced that having scale parameters for each semantically coherent region of an image is very intuitive.

---

> ### Author Rebuttal · Authors · 2025-07-31
>
> We thank all reviewers for their time and constructive feedback. We are encouraged by the generally positive comments, including:
> - This paper addresses a valuable and challenging problem of sparse, in-the-wild reconstruction (PYHN).
> - Our approach is technically sound (PYHN). Specifically, the semantic depth alignment is compelling (P8xk), and the multi-view geometry-guided supervision is interesting (B6Ss) and clever (P8xk, sq1N).
> - The experiments are comprehensive (sq1N, P8xk, PYHN) and consistent with the claims (B6Ss), demonstrating the effectiveness of our approach with clear improvements (PYHN, sq1N, P8xk).
> - Overall, the paper is well written and easy to follow (PYHN, B6Ss).
>
> We will also fix the typo that the limitations were discussed in Section 5 instead of 6, as noted by reviewers P8xk and B6Ss.
>
> Please find our responses to the reviewer' specific questions and comments below.
>
> ---
>
> (1) *Variance of improvements*
>
> We note that in Table 1, the pixel loss and feature loss each bring less improvement in LPIPS and DSIM than PSNR and SSIM. By adding additional constraints to multi-view optimization, we are trading off the ability to remove artifacts and the risk of smoothing the geometry. Since perceptual quality measures like LPIPS and DSIM are sensitive to high-frequency content, noisy/artifact images can get a boost in these metrics. We don't find significant variance in our experiments. Variance across three runs are 0.0045, 1.35e-5, 5.76e-5, 6.0e-6 for PSNR, SSIM, LPIPS, and DSIM respectively.
>
> ---
>
> (2) *Semantic scaling vs. single scaling*
>
> Please find the ablation study of semantic scaling vs. single image-level scaling in Section A.1.2 (Initialization Comparisons) in our appendix, which includes point cloud visualization, quantitative and qualitative comparisons. For convenience, a partial table is pasted below. The metrics are reported as average on the sparse MipNeRF 360 dataset.
>
> | Method                    | PSNR ↑ | SSIM ↑ | LPIPS ↓ | DSIM ↓ |
> |:--------------------------|:------:|:------:|:--------:|:------:|
> | image-level alignment     | 21.06  | 0.631  | 0.253    | 0.094  |
> | semantic alignment        | 21.96  | 0.690  | 0.216    | 0.080  |
>
> ---
>
> (Q1) *Semantic scaling validation*
>
> Thank you for your suggestion. We perform the recommended validation on the bonsai scene. Specifically, we use the Multi-View Stereo depth from dense views as GT depth estimates. A single per-frame estimate of scale results in a mean absolute error of 1.94, and our per‑semantic‑region scaling reduces this to 1.07. We also visualize the error map, which shows that the depth continuities around object boundaries exhibit higher error, while the depth inside semantic objects is more coherent and accurate after alignment. We attribute this to depth ambiguities from monocular depth that our piece-wise semantic alignment effectively mitigates. We will add this analysis and visualization to the revised manuscript.
>
> ---
>
> (Q2) "Clarification about virtual view regularization"
>
> Thank you for raising this point. In standard 3DGS optimization, training image pixels are only supervised along the rays that originate from their own camera center. With sparse views, this leaves large regions of the scene under-constrained. Introducing virtual cameras allows re-using the relatively accurate training views results and thereby enhancing multi-view consistency for constraint. For example, the color of a well-rendered training pixel is back-projected with its reasonably accurate depth and re-projected into the virtual view. If a "floater" Gaussian lies somewhere along the newly created ray at the virtual camera, it will alter the rendered color at the re-projected pixel, producing a photometric error. Minimizing this multi-view error as self-regularization forces the optimizer to suppress such floaters or inconsistencies.

---

> > ### Comment · Reviewer_B6Ss · 2025-08-05
> >
> > Thank you for the rebuttal. I believe the paper should be accepted and maintain my positive review.

---

### Official Review · Reviewer_P8xk · 2025-07-01

**Clarity:** 2
**Significance:** 3
**Originality:** 3
**Rating:** 4
**Confidence:** 4

**Summary:**

This paper presents MS-GS, a novel method that extends 3DGS for novel view synthesis under multi-appearance sparse-view conditions. The method offers two core contributions: (1) the Semantic Depth Alignment strategy that produces dense point cloud initialization using monocular depth aligned with SfM, guided by region-wise masks obtained from SAM; and (2) the Multi-view Geometry-guided Supervision that introduces virtual views and enforces pixel- and feature-level consistency via 3D warping. The authors also introduce a new benchmark dataset and an in-the-wild evaluation setting. Extensive experiments across multiple challenging datasets demonstrate the superior performance of MS-GS.

**Questions:**

1. The method relies on monocular depth estimation for initialization. How well does the proposed Semantic Depth Alignment generalize across different depth models (e.g., Depth Anything v2)? Does the method consistently outperform image-level alignment regardless of the chosen model?
2. The geometry-guided supervision may not generalize well to non-Lambertian scenes with reflections, shadows, or occlusions. Could the authors elaborate on how MS-GS handles such cases? Are there any specific failure cases where the 3D warping introduces artifacts or degrades performance?
3. Could the authors provide qualitative or quantitative examples where MS-GS struggles or fails (e.g., due to noisy segmentation or limited view overlap)? Such discussion would help readers better understand the practical boundaries of the method.

Additionally, I recommend including more visualization results in the main paper or supplementary material, such as qualitative comparisons across more scenes and a visual overview of the proposed dataset, to help readers gain a more intuitive understanding of the method and its effectiveness.

**Ethical Concerns:**

["NO or VERY MINOR ethics concerns only"]

**Final Justification:**

The authors have addressed most of my concerns, including the explanation of the coordinate alignment, the generalization to different depth models, additional visualization results, and error corrections. After considering other reviews and responses, I'd like to keep my original positive score.

**Limitations:**

Yes. The paper does acknowledge some limitations in Section 5, particularly regarding transient scene modeling. However, the discussion is brief and could be expanded. Potential limitations (e.g., sensitivity to segmentation quality or reflective materials) are not discussed and would be worth including in future revisions.

**Paper Formatting Concerns:**

No major formatting issues were observed. However, some minor citation errors and ambiguities (e.g., Table 2 and Equation 9) should be carefully reviewed and corrected.

**Quality:**

3

**Strengths And Weaknesses:**

**Strengths:**

1. The proposed approach is straightforward and effective, yielding clear improvements over baselines.
2. The Semantic Depth Alignment approach is novel and compelling. Its region-wise alignment strategy yields semantically meaningful point clouds that significantly outperform both sparse SfM and global alignment baselines.
3. The use of virtual views for multi-view supervision is clever and enhances consistency under sparse inputs.
4. The experiments are comprehensive, and the ablation studies are informative. The results consistently support the claimed improvements.

**Weaknesses:**

1. Although the authors claim to discuss limitations in Section 6, no such section exists. Only a few sentences in Section 5 briefly mention this topic.
2. As acknowledged, MS-GS does not currently handle transient or dynamic scene elements (e.g., vehicles and pedestrians), which limits its applicability in unconstrained real-world environments.
3. The effectiveness of geometry-guided supervision may degrade in cases of occlusion, reflections, or view-dependent effects (e.g., shadows or specularities). Further clarification on how the method handles these cases would be valuable.
4. Section A.5 could benefit from a clearer explanation. For instance, Equation (9) defines $P_{cam}$ as a $4 \times 3$ matrix but appears to represent a set of four 3D points. Clarifying the coordinate alignment would improve accessibility.
5. There are minor errors (e.g., an incorrect citation for FSGS in Table 2) that merit proofreading.

---

> ### Author Rebuttal · Authors · 2025-07-31
>
> We thank all reviewers for their time and constructive feedback. We are encouraged by the generally positive comments, including:
> - This paper addresses a valuable and challenging problem of sparse, in-the-wild reconstruction (PYHN).
> - Our approach is technically sound (PYHN). Specifically, the semantic depth alignment is compelling (P8xk), and the multi-view geometry-guided supervision is interesting (B6Ss) and clever (P8xk, sq1N).
> - The experiments are comprehensive (sq1N, P8xk, PYHN) and consistent with the claims (B6Ss), demonstrating the effectiveness of our approach with clear improvements (PYHN, sq1N, P8xk).
> - Overall, the paper is well written and easy to follow (PYHN, B6Ss).
>
> We will also fix the typo that the limitations were discussed in Section 5 instead of 6, as noted by reviewers P8xk and B6Ss.
>
> Please find our responses to the reviewer' specific questions and comments below.
>
> ---
>
> (2) *More discussion on limitation of MS-GS*
>
> Handling transient objects in 3D reconstruction under sparse-view conditions is severely under-constrained. While recent methods leverage uncertainty masks to remove transients and allow other observations to fill in the blank, oftentimes, no other observations exist under a sparse setting; therefore, the transient regions remain under-constrained regardless. In our experience, adding a masking mechanism to handle transients often leads to worse results.
>
> ---
>
> (3), (Q2), (Q3) *Non-Lambertian scenes and examples where MS-GS struggles*
>
> Generalizing to non‑Lambertian surfaces remains difficult even with dense views. Although we mitigate occlusion by comparing rendered and warped depths, the resulting mask can be imperfect, and specular highlights are smoothed or averaged out, as seen in our examples. It would be interesting to analyze how different techniques, such as explicitly modeling the light [1] and surface reconstruction [2], can be combined with our framework to further improve in‑the‑wild geometry reconstruction under sparse‑view conditions. More complex modeling requires more observations, which will further exacerbate the sparsity issue. This work mainly focuses on generalized appearance modeling.
>
> In addition, because MS-GS favors more accurate semantic regions for alignment, pixels that lie outside the semantic masks or are missed by imperfect segmentations lack dense initialization. Examples show that the empty regions can manifest as artifacts without geometric support.
>
> [1] Sun, Jia-Mu, et al. SOL-NeRF: Sunlight modeling for outdoor scene decomposition and relighting. SIGGRAPH Asia 2023 Conference Papers. 2023.
>
> [2] Huang, Binbin, et al. 2d Gaussian splatting for geometrically accurate radiance fields. ACM SIGGRAPH 2024 conference papers. 2024.
>
> ---
>
> (4) *Clearer explanation of the coordinate alignment*
>
> Thank you for pointing this out. We agree that the notation around Eq. (9) was ambiguous and will revise it. We intend $P_{\text{cam}}\in\mathbb{R}^{4\times3}$ to be a stack of four 3-D points per camera (one camera center and three points sampled one scaled unit $s$ along the unit rotation-axis directions), where each row is a point.  By incorporating these additional rotational points, we achieve more accurate camera alignment in both rotation and translation.
>
> The goal of coordinate alignment / in-the-wild setup is to disentangle training and test views in registration. Conventionally, they are registered with one run of Structure-from-Motion (SfM) so that test images are involved in the calibration, which leads to unrealistically better registration and denser initialization. In our setup, train views are first registered in one run, resulting in the coordinate system $C_1$. A separate run of SfM on train and test views results in the coordinate system $C_2$ because SfM reconstructs the scene in a different coordinate system each time. We compute a transformation between the shared train views, which are used to transform the test view from $C_2$ to $C_1$. In this way, we simulate the real-world scenario, where camera poses and 3D points in $C_1$ are estimated from only train views while having the test camera poses in the same coordinate system for evaluation.
>
> ---
>
> (5)  *Minor errors*
>
> Thank you for pointing this out. We will fix it and perform a more rigorous proofreading of the revised manuscript.
>
> ---
>
> (Q1) *Generalization to different depth models*
>
> In addition to Marigold [1], we have conducted an analysis of the semantic depth alignment vs. image-level alignment with Depth Anything v2 [2] and MiDaS 3.1 [3] on the sparse MipNeRF360 dataset. The metrics are reported as the average. As shown in the table below, our semantic depth alignment approach consistently improves the results, confirming its generalizability to different depth estimators.
>
> | Method                    | PSNR ↑ | SSIM ↑ | LPIPS ↓ | DSIM ↓ |
> |:--------------------------|:------:|:------:|:--------:|:------:|
> | **Marigold [1]**          |        |        |          |        |
> | image-level alignment     | 21.06  | 0.631  | 0.253    | 0.094  |
> | semantic alignment        | 21.96  | 0.690  | 0.216    | 0.080  |
> | **Depth Anything v2 [2]** |        |        |          |        |
> | image-level alignment     | 20.92  | 0.647  | 0.250    | 0.103  |
> | semantic alignment        | 21.99  | 0.686  | 0.214    | 0.077  |
> | **MiDaS 3.1 [3]**         |        |        |          |        |
> | image-level alignment     | 20.86  | 0.646  | 0.254    | 0.107  |
> | semantic alignment        | 21.82  | 0.681  | 0.222    | 0.084  |
>
> [1] Ke, Bingxin, et al. Repurposing diffusion-based image generators for monocular depth estimation. Proceedings of the IEEE/CVF conference on computer vision and pattern recognition. 2024.
>
> [2] Yang, Lihe, et al. Depth anything v2. Advances in Neural Information Processing Systems 37 (2024): 21875-21911.
>
> [3] Ranftl, René, et al. Towards robust monocular depth estimation: Mixing datasets for zero-shot cross-dataset transfer. IEEE transactions on pattern analysis and machine intelligence 44.3 (2020): 1623-1637.
>
>  ---
>
> *More visualization results*
>
> Thank you for your suggestion. We agree that more visualization results can assist in understanding of the method and its effectiveness. We have prepared more qualitative comparisons across scenes to demonstrate that MS-GS achieves better appearance rendering compared to baselines. We will add the results and a visual overview of the proposed dataset to the revised paper.

---

> > ### Comment · Reviewer_P8xk · 2025-08-05
> >
> > Thank you for the rebuttal, and most of my concerns have been addressed. I'd like to keep my original positive score.

---

### Official Review · Reviewer_sq1N · 2025-07-02

**Clarity:** 3
**Significance:** 3
**Originality:** 2
**Rating:** 4
**Confidence:** 4

**Summary:**

This paper introduces MS-GS, an approach for multi-appearance sparse-view 3DGS. MS-GS mainly consists of two components: semantic depth alignment and multi-view geometry-guided supervision. The former is proposed to address the problem of sparse SfM points in initialization, where the mono-depth estimates from an off-the-shelf model are aligned to the depths from SfM points. To perform more accurate alignment, MS-GS leverages SAM to identify semantically consistent regions and performs local alignment. Multi-view geometry-guided supervision is designed to provide additional training signals for the in-the-wild 3DGS setup. Specifically, training views are warped to novel views to create pseudo ground truth, which offers supervision for both rendered pixels and image features. Given 20 input views, the proposed method outperforms previous in-the-wild 3DGS methods, and ablation studies verify the effectiveness of the proposed method.

**Questions:**

1. The reviewer does not fully understand the proposed “in-the-wild” setup and the coordinate alignment approach. In this setup, is pose estimation performed separately for the training and test views? It would be helpful if the authors could further elaborate on the motivation and technical details of this design.

**Ethical Concerns:**

["NO or VERY MINOR ethics concerns only"]

**Final Justification:**

The proposed method outperforms the MASt3R counterpart from CVPR 2025 (which may be regarded as a concurrent work rather than a strong baseline). The authors also demonstrate that the method performs well under a sparser setup with 12 views, compared to their original experiments. These results address my concerns, and I have decided to raise my score to 4.

**Limitations:**

The authors briefly mention that the proposed method cannot handle transient objects but do not address the potential societal impact of this limitation.

**Paper Formatting Concerns:**

No.

**Quality:**

3

**Strengths And Weaknesses:**

Strengths:
1. Leveraging input views and the learned geometry to create virtual views to mitigate overfitting in the in-the-wild setup is a clever idea.
2. The proposed method outperforms baseline methods on multiple benchmarks. The experiments in this paper are comprehensive.
3. The ablation studies clearly verify the effectiveness of the proposed method.

Weaknesses:
1. Lack of novelty: Densifying initial SfM points for 3DGS and using semantic segmentation to identify semantically consistent regions for alignment have been explored in [1]. Although the technical details differ (for example, [1] uses MASt3R 3D points), the motivation and high-level idea of leveraging semantic segmentation are similar.
2. The authors use 20 input images for training, but 20 is not truly a “sparse” setting. Could the authors provide results (comparisons and ablations) with fewer input views?

[1] Tang et al., SPARS3R: Semantic Prior Alignment and Regularization for Sparse 3D Reconstruction, CVPR 2025.

---

> ### Author Rebuttal · Authors · 2025-07-31
>
> We thank all reviewers for their time and constructive feedback. We are encouraged by the generally positive comments, including:
> - This paper addresses a valuable and challenging problem of sparse, in-the-wild reconstruction (PYHN).
> - Our approach is technically sound (PYHN). Specifically, the semantic depth alignment is compelling (P8xk), and the multi-view geometry-guided supervision is interesting (B6Ss) and clever (P8xk, sq1N).
> - The experiments are comprehensive (sq1N, P8xk, PYHN) and consistent with the claims (B6Ss), demonstrating the effectiveness of our approach with clear improvements (PYHN, sq1N, P8xk).
> - Overall, the paper is well written and easy to follow (PYHN, B6Ss).
>
> We will also fix the typo that the limitations were discussed in Section 5 instead of 6, as noted by reviewers P8xk and B6Ss.
>
> Please find our responses to the reviewer' specific questions and comments below.
>
> ---
>
> (1) *Comparison with SPARS3R*
>
>  We thank the reviewer for referencing SPARS3R, which is a recent concurrent work that only focuses on *single-appearance* sparse reconstruction. Compared to SPARS3R, we propose to use monocular depth vs. two-view point map for initialization. We note that 1. our approach is more scalable and enables high-resolution depth, whereas MAST3R [1] can go OOM due to its quadratic pairwise bottleneck and produces low-resolution depth. 2. Monocular depth estimation, e.g., from Marigold [2], is much sharper, while the prediction from MAST3R is coarse and smooth, as MAST3R needs to meet two-view constraints. 3. MS-GS also incorporates virtual view supervision to regularize the scene geometry and appearance under the sparse-view setting.
>
> We also include the results of comparisons between MS-GS and SPARS3R below. The metrics are reported as the average on the sparse MipNeRF360 dataset.
>
> | Method                              | PSNR ↑ | SSIM ↑ | DSIM ↓ |
> |:------------------------------------|:------:|:------:|:------:|
> | SPARS3R                             | 21.48  | 0.674  | 0.081  |
> | MS-GS (No Virtual View Regularization) | 21.96  | 0.690  | 0.080  |
> | MS-GS                               | **22.39**  | **0.702**  | **0.072**  |
>
> We note that the result of MS-GS reported in Table 2 in our main manuscript is without virtual view regularization. Our dense initialization approach outperforms SPARS3R in PSNR and SSIM, demonstrating the advantage in using monocular depth estimation. Our virtual view regularization further improves the result.
>
> [1] Leroy, Vincent, Yohann Cabon, and Jérôme Revaud. Grounding image matching in 3d with mast3r. European Conference on Computer Vision. Cham: Springer Nature Switzerland, 2024.
>
> [2] Bingxin Ke, Anton Obukhov, Shengyu Huang, Nando Metzger, Rodrigo Caye Daudt, and Konrad Schindler. Repurposing diffusion-based image generators for monocular depth estimation. In Proceedings of the IEEE/CVF Conference on Computer Vision and Pattern Recognition, pages 9492–9502, 2024.
>
> ---
>
> (2) *Sparser setting*
>
> We note that in-the-wild reconstruction with sparse inputs and various appearances is a very challenging problem. Compared to the single appearance setup, more data is needed to disentangle different appearances and reconstruct a holistic scene. As for the Phototourism dataset, we use 2.62\%, 2.41\%, and 1.18\% of images compared to the original setting for "Brandenburg Gate", "Sacre Coeur", and "Trevi Fountain" scenes, respectively. The problem worsens as our newly introduced drone dataset represents 360-degree scenes, where the cameras are more divergent and have limited overlaps.
>
> To demonstrate the effectiveness of our methods in sparser settings, we conduct the experiments in the 12-view setting for the sparse drone dataset, where each appearance has 3 images. The metrics are reported as the average over the scenes. As shown in the table below, MS-GS continues to outperform the SoTA methods, and the performance gap widens compared to the 20-view setting. We will add these experiments and results to the revised manuscript.
>
> | Method            | PSNR ↑  | SSIM ↑ | LPIPS ↓ | DSIM ↓ |
> |:------------------|:-------:|:------:|:-------:|:------:|
> | GS-W              | 14.83   | 0.371  | 0.560   | 0.457  |
> | Wild-GS           | 13.66   | 0.289  | 0.587   | 0.583  |
> | WildGaussians     | 12.47   | 0.278  | 0.612   | 0.664  |
> | **MS-GS (ours)**  | **17.78** | **0.477** | **0.412** | **0.180** |
>
> ---
>
> (Q1) *In this setup, is pose estimation performed separately for the training and test views?*
>
> Yes, the train and test views are registered separately and aligned together. Conventionally, they are registered with one run of Structure-from-Motion (SfM) so that test images are involved in the calibration, which leads to unrealistically better registration and denser initialization. In our setup, train views are first registered in one run, resulting in the coordinate system $C_1$. A separate run of SfM on train and test views results in the coordinate system $C_2$ because SfM reconstructs the scene in a different coordinate system each time. We compute a transformation between the shared train views, which are used to transform the test view from $C_2$ to $C_1$. In this way, we simulate the real-world scenario, where camera poses and 3D points in $C_1$ are estimated from only train views while having the test camera poses in the same coordinate system for evaluation.

---

> > ### Comment · Reviewer_sq1N · 2025-08-03
> >
> > I appreciate the authors’ efforts during the rebuttal. I am glad to see that the proposed method outperforms the MASt3R counterpart and remains effective under sparser (12-view) settings. I also believe monodepth is particularly suitable for this task, as it avoids the multi-view inconsistency introduced by appearance variations.
> >
> > Since the authors have addressed most of my concerns, I would like to raise my score to 4.

---

> > > ### Author Response · Authors · 2025-08-04
> > >
> > > We appreciate the reviewer's thoughtful comments and discussion. We are glad that your concerns are mostly resolved, hence the raised score.
> > >
> > > Thank you again for your time and constructive feedback that improved the paper.

---

### Official Review · Reviewer_PYHN · 2025-07-06

**Clarity:** 3
**Significance:** 3
**Originality:** 3
**Rating:** 5
**Confidence:** 4

**Summary:**

## Paper Summary
This paper addresses an important problem: novel view synthesis (NVS) in the wild from a sparse set of input images. It introduces two key technical contributions along with a new dataset.

The two technical contributions are:
1. Semantic depth alignment: This step leverages Segment Anything (SAM) and structure-from-motion (SfM) points to correct distortions in monocular depth predictions. These aligned points are then used to initialize the proposed NVS-in-the-wild model.
2. Multi-view geometry-guided supervision: This component enforces both photometric and feature consistency. The losses are computed using rendered virtual views and point-projected virtual views.

The dataset contribution is a large-scale, unbounded drone dataset captured under diverse weather and lighting conditions.

Evaluation: The method is evaluated on three datasets: two standard benchmarks (Mip360 and Phototourism) and the newly proposed drone dataset.

## Review Summary
The technical contributions are solid and directly address the challenge of sparse-view NVS in the wild. The evaluation is thorough and demonstrates the effectiveness of the proposed approach. The primary weakness lies in the limited details provided about the newly introduced dataset.

**Questions:**

1. Details of the proposed dataset? And how does it address the limitation of requiring half of a test image in the previous evaluation pipeline?
2. Runtime and memory analysis?

**Ethical Concerns:**

["NO or VERY MINOR ethics concerns only"]

**Final Justification:**

The author's response has addressed my concerns, and I'd like to keep my original positive score.

**Limitations:**

Limitations are not discussed.

**Quality:**

3

**Strengths And Weaknesses:**

## Strengths
- The paper is very well written and easy to follow.
- It addresses a valuable and challenging problem: NVS in the wild from sparse input views.
- The proposed technical contributions are well motivated and technically sound.

## Weaknesses
1. The main concern is the lack of clarity around the newly introduced dataset. Lines 78–86 describe the motivation for a new dataset to enable NVS-in-the-wild evaluation without requiring access to half of the test images, as was common in previous benchmarks. However, the dataset is only briefly described in Lines 232–237. It remains unclear how the proposed dataset addresses the limitations of prior setups, specifically the need to reserve half of the test images.
2. The paper lacks a runtime and memory analysis. Given that the proposed method is based on 3D Gaussian Splatting (3DGS) and uses dense initialization points, it is important to quantify the additional memory usage and training time compared to a more traditional sparse-point initialization.

---

> ### Author Rebuttal · Authors · 2025-07-31
>
> We thank all reviewers for their time and constructive feedback. We are encouraged by the generally positive comments, including:
> - This paper addresses a valuable and challenging problem of sparse, in-the-wild reconstruction (PYHN).
> - Our approach is technically sound (PYHN). Specifically, the semantic depth alignment is compelling (P8xk), and the multi-view geometry-guided supervision is interesting (B6Ss) and clever (P8xk, sq1N).
> - The experiments are comprehensive (sq1N, P8xk, PYHN) and consistent with the claims (B6Ss), demonstrating the effectiveness of our approach with clear improvements (PYHN, sq1N, P8xk).
> - Overall, the paper is well written and easy to follow (PYHN, B6Ss).
>
> We will also fix the typo that the limitations were discussed in Section 5 instead of 6, as noted by reviewers P8xk and B6Ss.
>
> Please find our responses to the reviewer' specific questions and comments below.
>
> ---
>
> (Q1) *More clarity on the newly introduced dataset*
>
> To obtain the appearance embeddings of a test image from the Phototourism [1] dataset, prior setups either optimize on half of the test image or use a network to extract features from the entire test image. Both of these approaches lead to the utilization of test images during evaluation. For our dataset, the multi-view, multi-appearance setup allows the correct appearance to be rendered based on metadata/timestamp, not pixel-wise information from the test image. For example, a snowy test image can be rendered with the appearance embedding of a snowy training image by relating their timestamps. This setup is also faster than optimizing half of the test image. Additionally, our dataset contains scenes with 360-degree coverage by perspective cameras, whereas Phototourism is only covered by face-forward images. We will include this clarification in the revised manuscript.
>
> [1] Noah Snavely, Steven M Seitz, and Richard Szeliski. Photo tourism: exploring photo collections
> in 3d. In ACM siggraph 2006 papers, pages 835–846. 2006.
>
> ---
>
> (Q2) *Runtime and memory analysis*
>
> Thank you for your suggestion. We report the runtime and peak GPU memory usage of each scene in the table below.
>
> |**Runtime (min) / GPU memory (GB)**| Bicycle   | Bonsai    | Counter   | Garden    | Kitchen   | Room      | Stump     | Average   |
> |--------------|-----------|-----------|-----------|-----------|-----------|-----------|-----------|-----------|
> | Sparse       | 14.4/14.6 | 7.8/14.5  | 7.8/14.5  | 13.7/14.6 | 8.7/14.6  | 7.7/14.6  | 11.7/14.5 | 10.2/14.6 |
> | Dense init.  | 16.7/16.2 | 9.5/15.6  | 9.1/15.4  | 15.8/16.5 | 10.0/15.6 | 9.0/15.6  | 13.1/15.8 | 11.9/15.8 |
>
> The training time for 15,000 iterations is increased by 1.7 minutes on average with dense initialization compared to sparse initialization. While the optimization of more points is more costly, the pruning process can quickly remove most of redundant dense initialization. The peak GPU memory usage is increased by 1.2 GB on average. With our dense initialization, 3DGS converges faster to the final number of Gaussians and photometric loss.

---

> > ### Comment · Reviewer_PYHN · 2025-08-05
> >
> > The author's response has addressed my concerns, and I'd like to keep my original positive score.

---

### Note · Authors · 2025-08-12

We sincerely thank all reviewers for their constructive feedback and thoughtful discussion that improved the paper. We are encouraged by the positive comments that:

* This paper addresses a valuable and challenging problem of sparse, in-the-wild reconstruction (PYHN).
* Our approach is technically sound (PYHN). Specifically, the semantic depth alignment is compelling (P8xk), and the multi-view geometry-guided supervision is interesting (B6Ss) and clever (P8xk, sq1N).
* The experiments are comprehensive (sq1N, P8xk, PYHN) and consistent with the claims (B6Ss), demonstrating the effectiveness of our approach with clear improvements (PYHN, sq1N, P8xk).
* Overall, the paper is well written and easy to follow (PYHN, B6Ss).

We are glad that critical concerns from all reviewers have been addressed and reached a positive consensus. We will incorporate the new results, analysis, and discussion into the revised paper.

Thank you again for your valuable time and feedback.

---

### Decision · Program_Chairs · 2025-09-17

**Decision:**

Accept (poster)

**Comment:**

The paper received positive reviews with scores of 4, 4, 4, and 5 (mean: 4.25). Reviewers highlighted the significance of tackling sparse-view novel view synthesis in the wild (PYHN, sq1N), the effectiveness of semantic depth alignment (P8xk), multi-view geometry-guided supervision (sq1N, B6Ss), and the comprehensive experiments and clear presentation (PYHN, P8xk, B6Ss). Results consistently surpass baselines, with ablations supporting the key contributions.

Concerns centered on limited dataset description (PYHN), overlap with SPARS3R (sq1N), and insufficient discussion of limitations such as transient objects and non-Lambertian surfaces (P8xk, B6Ss). The rebuttal provided clarifications, additional experiments, and analyses, which reviewers agreed resolved their main issues.

After carefully considering the reviews and rebuttal, the AC concurs with the reviewers’ consensus and recommends the paper for **acceptance**.

For the camera-ready version, the authors should ensure all discussions from the rebuttal are incorporated into the main paper and supplementary materials. The specific changes that need to be implemented are:

**1. Dataset clarity (PYHN)**: Provide a fuller description of the new dataset and why it avoids test-image leakage compared to prior setups.

**2. Relation to prior work (sq1N)** : Clearly position MS-GS relative to SPARS3R and articulate the novelty.

**3. Limitations and edge cases (P8xk, B6Ss)**: Expand discussion and examples on transient/dynamic content, reflections, occlusions, and segmentation errors.

**4. Presentation (P8xk)**: Correct minor errors (citations, Table 2 and Equation 9) and add more visualizations and a dataset overview.

**5. Semantic scaling (B6Ss)**: Include analysis and visualization for semantic scaling validation.